# Sex differences in the risk profiles for anemia in people living with HIV, A cross sectional study

Kingsley Kamvuma[1,2]*, Benson M. Hamooya[3], Kaseya O. R. Chiyenu[4], Yusuf Uthman Ademola[1], Steward Mudenda[5], Alfred Machiko[6], Sepiso K. Masenga[1], Sody M. Munsaka[2]

**1** Department of Pathology and Microbiology, HAND Research Group, School of Medicine and Health Sciences, Mulungush University, Livingstone Campus, Livingstone, Zambia, **2** Department of Biomedical Sciences, School of Health Sciences, University of Zambia, Lusaka, Zambia, **3** Department of Public Health, HAND Research Group, School of Medicine and Health Sciences, Mulungush University, Livingstone Campus, Livingstone, Zambia, **4** Department of Internal Medicine, Livingsotne University Teaching Hospital, Livingstone, Zambia, **5** Department of Pharmacy, School of Health Sciences, University of Zambia, Lusaka, Zambia, **6** Department of Basic Science, School Medicine, Cobberbelt Unversity, Ndola, Zambia

* kamvumak@yahoo.com

## Abstract

### Background

Anemia in people living with HIV (PLWH) significantly impacts quality of life and health outcomes. This study aimed to determine sex differences in factors associated with anemia in PLWH at Livingstone University Teaching Hospital, Zambia.

### Methods

This cross-sectional study involved 631 PLWH aged 18 years or older who had been on combinational ART for at least 6 months. Data was collected via standardized questionnaires and medical records. Anemia was defined as haemoglobin levels < 13 g/dL for men and < 12 g/dL for women, based on WHO criteria. Logistic regression models assessed the associated factors, stratified by sex.

### Results

Participants had a median age of 44 years, with a female preponderance of 64.2%. The overall prevalence of anemia was 36%, significantly higher in females (41.1%) compared to males (27.2%) (p < 0.001). In females, waist circumference (AOR = 0.97, 95% CI: 0.95-0.99, P = 0.018), albumin levels (AOR = 0.96, 95% CI: 0.92-0.99, P = 0.047), NNRTI regimens (AOR = 2.78, 95% CI: 1.34-5.78, P = 0.006), microcytosis (AOR = 3.18, 95% CI: 1.26-8.03, P = 0.014), and hypertension (OR = 0.34, 95% CI: 0.13-0.87, P = 0.024) were linked to anemia in adjusted analysis but these associations were abrogated by male sex.

**Data availability statement:** All relevant data are within the manuscript and its Supporting Information files.

**Funding:** The author(s) received no specific funding for this work.;

**Competing interests:** The authors have declared that no competing interests exist.

## Conclusions

We found a 36% prevalence of anemia among PLWH, with a higher prevalence in females (41%) compared to males (27%), including severe forms of anemia. Among females, anemia was linked to lower waist circumference, lower albumin levels, NNRTI regimens, microcytosis, and blood pressure but not males. Further studies are warranted.

## Introduction

Anemia is a common comorbidity in people living with HIV (PLWH), significantly impacting their quality of life and clinical outcomes [1]. A recent meta-analysis revealed that the global pooled prevalence of anemia in adults for both men and non-pregnant women is approximately 46.6% [2]. However, the prevalence of anemia in PLWH is disproportionately higher in sub-Saharan Africa (SSA), particularly in Southern Africa (58 to 70%) [3–8]. This indicates that despite advancements in combinational antiretroviral therapy (ART), anemia remains prevalent in this population.

Anemia PLWH can exacerbate HIV-related complications, such as immunosuppression, opportunistic infection, and poor treatment outcomes including disease progression and death [9–11]. Aneamia in PLWH maybe a marker of underlying comorbidities, such as nutritional deficiencies, chronic inflammation, and other chronic diseases [10,12]. It can also worsen quality of life, leading to fatigue, weakness, and decreased productivity [13]. Therefore, there is a need to conduct routine screening of anemia and risk factors associated with anemia among PLWH [3].

HIV and anemia share a complex relationship driven by multifactorial mechanisms [14]. Chronic inflammation is a hallmark of HIV infection that persists even in the presence of ART [15]. The virus itself triggers a cascade of immune responses, leading to a continuous state of inflammation which disrupt erythropoiesis and increase iron sequestration[16]. Opportunistic infections and coexisting conditions, such as tuberculosis, further exacerbate anemia. Additionally, certain antiretroviral therapy (ART) regimens, particularly those containing zidovudine, are known to suppress bone marrow function [17]. HIV can also impair the survival/proliferative capacity of hematopoietic progenitor cells which can further suppress erythropoiesis [18]. Nutritional deficiencies, including iron, folate, and vitamin B12 deficiencies, are also prevalent in people living with HIV (PLWH) [19].

Existing studies have largely focused on the general population, neglecting the unique needs and sex specific characteristics of PLWH [2,9]. While it is known that females may be at a higher risk of developing anemia due to various biological factors [13,20], and that men may also experience anemia due to various factors non specific factors, there is still little or no information available about sex differences among PLWH[21]. Investigating sex differences in anemia among PLWH is cardinall as biological, hormonal, and social factors uniquely influence anemia risk in men and women [22]. Moreover, ART-related myelotoxicity and access to healthcare can differ by sex, further influencing anemia outcomes [23]. Understanding these differences is vital for develop gender specific interventions, optimizing anemia management strategies, and addressing gender specific disparities in health outcomes among PLWH.

This study aims to provide an understanding of the sex specific risk profiles for anemia among PLWH. The overall goal of the study was to identify sex-specific factors associated with anemia in PLWH attending an HIV clinic at Livingstone University Teaching Hospital (LUTH) in Zambia.

## Methods

### Study population and eligibility

This cross-sectional study was conducted among PLWH attending an ART clinic at LUTH, an urban tertiary care centre. Participants were recruited between 25[th] August 2023 and 30[th] December 2023. Following informed consent, we recruited adult PLWH aged 18 years and above who had been receiving ART for ≥ 6 months. The study focused on adults because anemia's risk factors and management differ significantly from those in children and adolescents due to developmental and physiological differences [24]. We recruited participants who had been on ART for at least six months to ensure that anemia outcomes were assessed after a sufficient period on treatment, reflecting the long-term effects or stabilization of hematologic and immunologic parameters [25]. Pregnant women, participants with excessive menstrual bleeding and disorders of haemoglobin synthesis including Sickle cell anemia and thalassaemia or malignant neoplasm were excluded.

### Data collection

Following informed consent data was collected using a structured questionnaire designed to capture demographic, clinical, and treatment-related information. These included questions on age, sex, ART regimen, duration on ART, and medical history. The questionnaires were administered during routine clinic visits, supplemented by data extraction from patient files and laboratory records to ensure comprehensive and accurate information. The questionnaire covered demographic details (age, gender, marital status, and education level), HIV-related specifics (duration of ART, ART regimen and viral load), presence of comorbidities (hypertension and hepatitis B), lifestyle factors (Excercise habits) and anthropometric measurements including Body Mass Index (BMI) and Waist circumference (WC). Waist circumference was measured using a flexible tape measure positioned at the midpoint between the iliac crest and the lower rib margin, with measurements recorded to the nearest 0.1 cm. Body mass index (BMI) was calculated as weight in kilograms divided by the square of height in meters (kg/m²).

We also collected laboratory samples for measurements of various parameters including viral load samples, full blood counts in ethylenediaminetetraacetic acid (EDTA) containers; a Becton Dickson flow cytometer was used to analyse CD4 counts and the viral load was analysed using Ampliprep/Taqman 96 PCR analyser in the laboratories at LUTH. Biochemical analyses were done on a Pentra C200 and HumaStar 80 clinical chemistry analyzer (Human Diagnostics, Germany) using kits supplied by the manufacturer.

Antiretroviral therapy regimens included non-nucleoside reverse transcriptase inhibitor (NNRTI) with options for efavirenz (EFV) or nevirapine (NVP) paired with abacavir/lamivudine or emtricitabine (ABC/3TC or FTC) or tenofovir disoproxil fumarate/emtricitabine (TDF/FTC). Protease inhibitor (PI) regimens included lopinavir/ritonavir (LPV/r) or atazanavir/ritonavir (ATV/r) combined with ABC/3TC or FTC, AZT/FTC, or TDF/FTC. Integrase strand transfer inhibitor (INSTI) regimens included dolutegravir (DTG) with TDF/3TC.

### Sample size determination

The sample size calculation utilized the single population proportion formula, assuming a proportion of 50%. We set the desired margin of error at 0.05 and a confidence level of 99%. With a total population of 3,880 patients receiving antiretroviral therapy (ART) at LUTH, sample size was determined to be 578. To accommodate potential ineligibility after consent and non-response, a 10% contingency was added. Therefore, the final sample size was calculated to be 631 participants.

## Operational definitions

Viral suppression was defined according to the Zambia Consolidated Guidelines for Treatment and Prevention of HIV Infection as consistent adherence to antiretroviral therapy (ART) and achieving a viral load of less than 1,000 copies of HIV RNA per milliliter of blood [26]. The primary outcome anemia was defined as haemoglobin < 13 g/dL for men and < 12 g/dL for women, according to the World Health Organization criteria [27]. Further subclassified the anemia as mild (11–12.9 g/dl for males and 11–11.9 g/dl for females), moderate (8–10.9/dl) and severe (<7.9 g/dl) [27]. The type of anemia s was assessed using mean cell volume (MCV) values, categorizing them as microcytosis (<80 fL), normocytosis (80-100 fL), and macrocytosis (>100 fL) [28]

## Statistical analysis

For our statistical analyses, we employed SPSS software (version 22). Categorical data were summarized using frequencies and proportions. Statistical significance for continuous variables was assessed using Two-way ANOVA with Fisher's Least Significant Difference multiple comparison test (*$p < 0.05$; **$p < 0.01$, ***$p < 0.001$; ns = $p > 0.05$). We further conducted logistic regression (both univariable and multivariable) to estimate factors associated with anemia, stratified by sex. The covariates included in the final models were selected based on published evidence and variables that demonstrated statistical significance in the univariable analysis. Additionally, we assessed model fitness using the Hosmer-Lemeshow goodness-of-fit test, with statistical significance defined as $P < 0.05$.

## Ethical approval

This study was approved by the University of Zambia Biomedical Research Ethics Committee (UNZABREC- REF. NO. 4062-2023) and all participants provided written informed consent. The study was conducted in accordance with the Declaration of Helsinki and Good Clinical Practice guidelines.

## Reporting format

We adhered to the strengthening the reporting of observational studies in epidemiology guidelines for reporting observational studies. See Supplementary 1 file.

# Results

## Background characteristics

We studied 631 PLWH with a median age of 44 years and 63.2% female predominance. We found the prevalence of anemia to be 36% overall but was higher in females compared to males (41% vs 27%, $p < 0.001$) (Fig 1). Among females The majority of participants were virally suppressed (81.6%) and were on NNRTI based regimen (67.2%) (Table 1). Among females, 21.1% had mild anemia, 15% had moderate anemia, and 5% had severe anemia. In contrast, 21.1% of males had mild anemia, 5.6% had moderate anemia, and only 0.4% had severe anemia (Fig 1).

## Relationship between anemia and sex specitic characteristics in PLWH

We found that male participants on NNRTI regimens were more likely to have anemia compared to those on INSTI or PI regimens ($p = 0.03$), (Table 1). Similarly, females on NNRTI regimens were more likely to be anemic compared to those on INSTI or PI

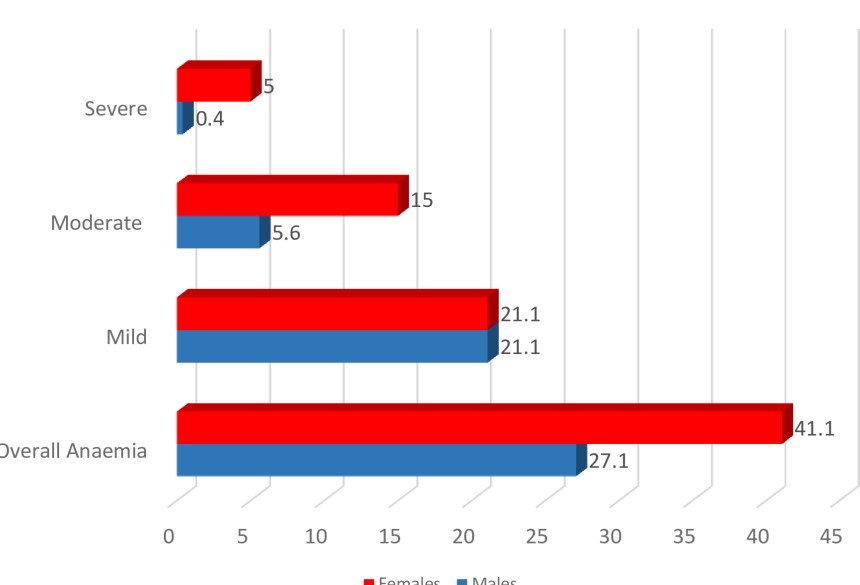

**Fig 1. Sex differences in prevalence of anemia and its severity.** Females exhibit a higher overall prevalence of anemia (41.1%) compared to males (27.1%). In terms of severity, both sexes have equal prevalence of mild anemia (21.1%). However, moderate anemia is more common in females (15%) than in males (5.6%), and severe anemia is significantly higher in females (5%) compared to males (0.4%).

regimens (p = 0.008). Hypertensive females were less likely to be anemic (25% vs 43.3%, P = 0.011), but this relationship was not observed in males. Based on the Mean Cell Volume (MCV), there were no significant differences across MCV categories in females. However, 57% of anemic females had microcytosis, compared to 39% with normocytosis and 39% with macrocytosis (p = 0.055). In contrast, anemic males were more likely to have macrocytosis (40%) compared to normocytosis (24%) and microcytosis (16%) (p = 0.035) (Table 1).

Females with anemia were significantly younger than their male counterparts (mean age 44.0 ± 11.5 vs. 48.5 ± 15.9 years, p < 0.05) (Fig 2A). Both males and females with anemia had lower body mass index (BMI) compared to those without anemia (males: 20.7 ± 3.5 vs. 22.2 ± 5.1 kg/m²; females: 23.9 ± 5.8 vs. 24.5 ± 5.1 kg/m², p < 0.0001) (Fig 2B).

Anemic males had a significantly lower waist circumference (WC) than both non-anemic males (75.6 ± 11.3 cm vs. 80.3 ± 12.8 cm) and anemic females (75.6 ± 11.3 cm vs. 81.1 ± 11.1 cm, p < 0.001). Similarly, anemic females had a lower WC compared to non-anemic females (81.1 ± 11.1 cm vs. 83.9 ± 12.9 cm). (Fig 2C). Furthermore, females with anemia had been on ART longer than those without anemia (124 ± 151 vs. 103 ± 50.1 months, p < 0.05) Fig 2D. Alanineaminotransferase levels were higher in males with anemia compared to females with anemia (24.8 ± 27.1 vs. 20.3 ± 10.4 mmol/l, p < 0.05) (Fig 2E). We did not find statistical sex differences between the anemic and non anemic groups in (Fig 2F) aspartate aminotransferase levels, (Fig 2G) asolute WBC count, (Fig 2H) neutrophil count, (Fig 2I) lymphocyte counts. Monocyte counts showed sex-specific differences, being lower in females with anemia but higher in females without anemia compared to males with or without anemia (males: 0.4 ± 0.2 vs. 0.5 ± 0.8 x10⁹/L vs female: 0.1 ± 0.01 vs. 0.3 ± 0.1 x10⁹/L, p < 0.01) (Fig 2J). Platelet counts were higher in females with anemia compared to females without anemia (276.0 ± 80 vs. 257.6 ± 80 x10⁹/L, p < 0.05) (Fig 2K). We also did not find statically significant

**Table 1. Shows SEX specific characteristics stratified by anemia status.**

| Background characteristics | Total | MALES | | P | FEMALES | | p |
|---|---|---|---|---|---|---|---|
| | | Anemia 63 (27%) | No anemia 169 (73%) | | Anemia 164 (41.1) | No anemia 235(58.9) | <0.001 |
| Physical Excerise | | | | 0.229 | | | 0.834 |
| *Yes* | 189 | 21(22.8) | 71(77.2) | | 39 (40.2) | 58(59.8) | |
| *no* | 437 | 42(30.0) | 98(70.0) | | 123 (41.4) | 174(58.6) | |
| Marital status | | | | 0.683 | | | 0.535 |
| *Yes* | 326 | 43(26.4) | 120 (73.6) | | 64 (39.3) | 99 (60.7) | |
| *No* | 305 | 20(28.9) | 49(71.0) | | 100 (42.4) | 136(57.6) | |
| ART regimen | | | | **0.03** | | | **0.008** |
| *INSTI* | 132 | 8(14.0) | 49(86.0) | | 20 (26.7) | 55(73.3) | |
| *NNRTI* | 424 | 47(32.4) | 98(67.6) | | 128(45.9) | 151(54.1) | |
| *PI* | 75 | 8(26.7) | 22(73.3) | | 16(35.6) | 29(64.4) | |
| Duration on ART mo, m(IQR) | 631 | 120 (48, 144) | 108(64,149) | 0.919 | 108(60, 144) | 108(60, 144) | 0.492 |
| HBsAg | | | | 0.350 | | | 0.657 |
| *Yes* | 42 | 4(40) | 6(60) | | 12(37.5) | 20(62.5) | |
| *No* | 588 | 59(26.6) | 163(73.4) | | 152(41.5) | 214(58.5) | |
| Viral load | | | | 0.748 | | | 0.451 |
| *Suppressed* | 514 | 51 (26.4) | 142 (73.6) | | 129 (40.2) | 192 (59.8) | |
| *Unsuppressed* | 116 | 11 (28.9) | 27 (71.1) | | 35 (44.9) | 43 (55.1) | |
| CD4 + Count | | | | 0.150 | | | 0.168 |
| *<200* | 35 | 3 (23.1) | 10 (76.9) | | 12 (54.5) | 10 (45.5) | |
| *201-350* | 49 | 4 (12.9) | 27 (87.1) | | 18 (32.1) | 38 (68.9) | |
| *>351* | 505 | 55 (29.4) | 132 (70.6) | | 133 (41.8) | 185 (58.2) | |
| Hypertension | | | | 0.125 | | | **0.011** |
| *Yes* | 79 | 4 (14.8) | 23 (85.2) | | 13 (25) | 39 (75) | |
| *No* | 552 | 59 (29.2) | 146 (70.8) | | 151 (43.3) | 196 (56.7) | |
| MCV | | | | **0.035** | | | 0.055 |
| *Microcytosis* | 77 | 4 (16.7) | 20 (83.3) | | 30 (56.6) | 23 (43.4) | |
| *Normocytosis* | 397 | 36 (24) | 114 (76) | | 97 (39.3) | 150 (60.7) | |
| *Macrocytosis* | 148 | 22 (40) | 33 (60) | | 36 (38.7) | 57 (61.3) | |

Abbreviations: m (IQR): Mean (Interquartile Range), BMI: Body Mass Index, WC: Waist Circumference, ALT: Alanine Transaminase, AST: Aspartate Transaminase, MCV: Mean Cell Volume, WBC: White Blood Cells, GFR: Glomerular Filtration Rate, ART: Antiretroviral Therapy, INSTI: Integrase Strand Transfer Inhibitors, NNRTI: Non-Nucleoside Reverse Transcriptase Inhibitors, PI: Protease Inhibitors

differences creatinine (Fig 2L) and eGFR values (Fig 2M) although females without anemia had lower values (males: 113 ± 19 vs 113 ± 21 vs female 107 ± 20 vs 101 ± 20) and urea levels (females: 4.53 ± 1.13 vs 3.47 ± 0.07 mg/dl vs males: 3.12 ± 0.14 vs 3.60 ± 0.10 mg/dl) (Fig 2N). Lastly, albumin levels were lower in both males and females with anemia compared to those without anemia (males: 39.8 ± 6.2 vs. 42.6 ± 7.9 kg/m²; females: 40.4 ± 6.8 vs. 42.0 ± 7.1 kg/m², p < 0.05) (Fig 2O).

## Factors associated with anemia in females

In the unadjusted analysis, several factors were significantly associated with anemia in females, (Table 2). The odds of anemia decreased by 2% with each additional year in age (OR = 0.98, 95% CI: 0.96-0.99, P = 0.03). Lower waist circumference was linked to a 2% increase in the odds of anemia (OR = 0.98, 95% CI: 0.96-0.99, P = 0.024). Increased monocyte levels

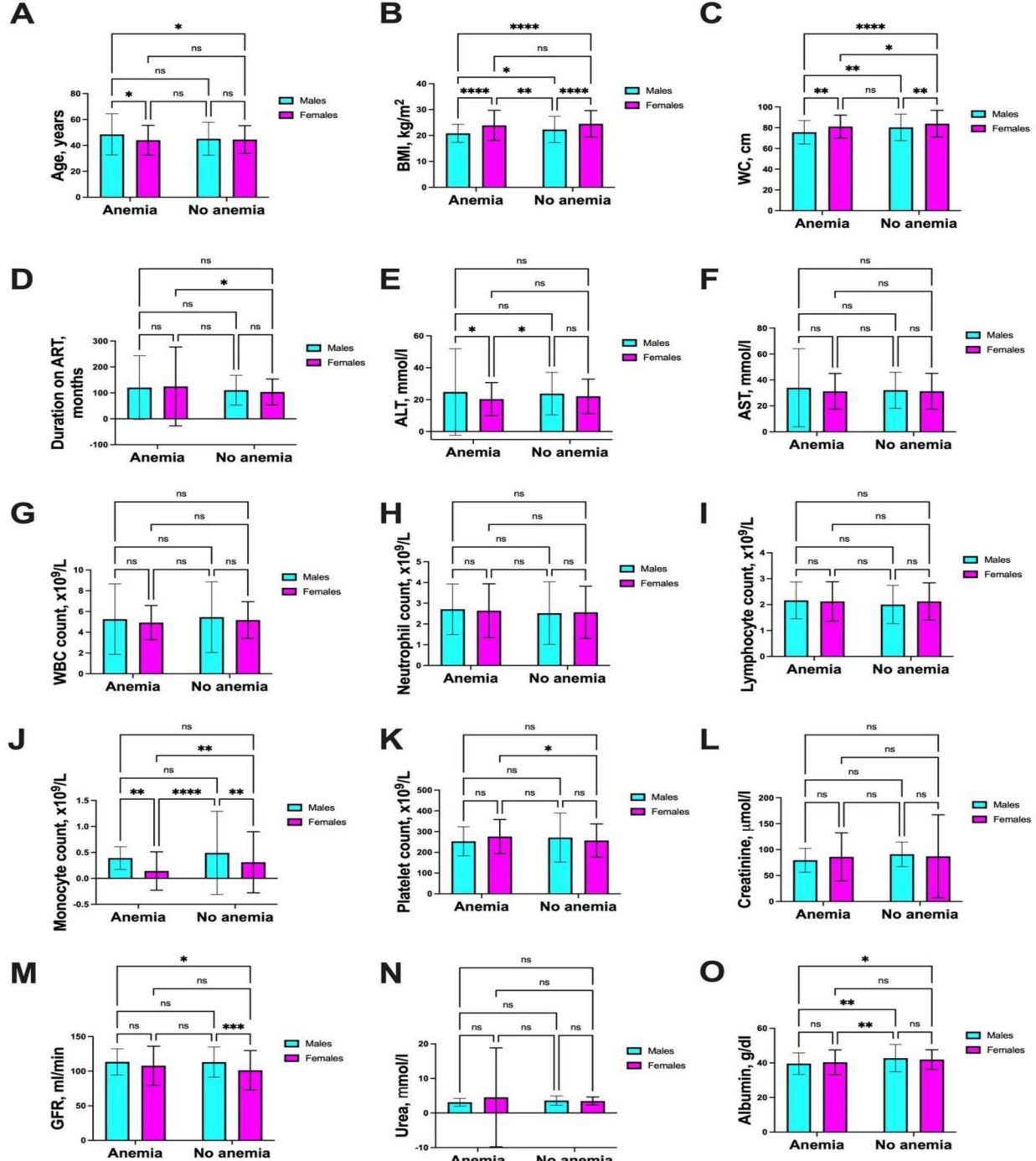

**Fig 2. Sex differences and comparison of characteristics in the study.** ) Shows differences in males and females with anemia compared to those without. (A) Age, (B) Body mass index, (C) waist circumference, (D) Duration on ART, (E) Alanine aminotransferase levels, (F) Aspatate aminotransferase, (G) White blood Counts counts, (H)neutrophil counts, (I) lymphocyte counts, (J)monocytes, (K)platelets, (L)creatinine, (M) eGFR, (N)urea, (O) albumin exhibit variations between groups. Statistical significance was assessed using Two-way ANOVA with Fisher's LSD multiple comparison test (*p < 0.05; **p < 0.01, ***p < 0.001; ns = p > 0.05).

**Table 2. Logistic regression analysis of factors associated with anemia in females.**

| Variables | Unadjusted analysis | | Adjusted Analysis | |
|---|---|---|---|---|
| | OR (95% CI) | *P* | AOR (95% CI | *P* |
| Age (years) | 0.98 (0.96,0.99) | **0.03** | 0.99 (0.97 -1.02) | 0.885 |
| Bmi kg/m2 | 0.98 (0.94,1.02) | 0.258 | | |
| WC (cm) | 0.98 (0.96,0.99) | **0.024** | 0.97 (0.95-0.99) | **0.018** |
| Duration on ART mo, m(IQR) | 1.00 (0.99,1.00) | 0.08 | 1.001 (0.99-1.01) | 0.402 |
| Art regimen | | | | |
| *INSTI* | 1 | | 1 | |
| *NNRTI* | 2.33(1.32,4.09) | **0.003** | 2.78 (1.33-5.82) | **0.006** |
| *PI* | 1.52(0.68, 3.37) | 0.305 | 2.58 (0.89-7.46) | 0.08 |
| Mean cell Volume | | | | |
| *Macrocytosis* | 1 | | 1 | |
| *Microcytosis* | 2.07 (1.04-4.09) | **0.038** | 3.18 (1.27-7.99) | **0.014** |
| *Normocytosis* | 1.02 (0.63-1.67) | 0.925 | 1.59 (0.81-3.13) | 0.183 |
| CD4 + Count | | | | |
| *>350* | 1 | | 1 | |
| *<200)* | 1.67 (0.70-3.98) | 0.247 | 3.08 (0.92-10.28) | 0.068 |
| *201-350* | 0.66 (0.36-1.21 | 0.175 | 0.45 (0.2-1.03) | 0.060 |
| HTN, mmHg | | | | |
| *No* | 1 | | 1 | |
| *Yes* | 0.43 (0.22-0.84) | **0.013** | 0.34 (0.14-0.87) | **0.024** |
| Viral Load | | | | |
| *Suppressed* | 1 | | 1 | |
| *Unsuppressed* | *0.83 (0.50-1.36)* | 0.451 | 1.03 (0.52-2.04) | 0.943 |
| Lymphocytes 10⁹/L | 0.97 (0.72, 1.30) | 0.847 | | |
| Monocytes 10⁹/L | 0.29 (0.09, 0.96) | **0.043** | 0.59 (0.15-2.27) | 0.441 |
| Creatinine μmol/l | 0.99 (0.99, 1.01) | 0.841 | | |
| Albumin g/dl | 0.96 (0.92, 0.99) | **0.036** | 0.96 (0.92-0.99) | **0.047** |

Abbreviations: BMI Body Mass Index, WC Waist Circumference, mo months, cm centre meters, mmHg millimetres of mercury, ART Antiretroviral therapy, CD4 cluster of differentiation 4, HTN Hypertension, INSTI: Integrase Strand Transfer Inhibitors, NNRTI: Non-Nucleoside Reverse Transcriptase Inhibitors, PI: Protease Inhibitors

were associated with a 71% decrease in the odds of anemia (OR = 0.29, 95% CI: 0.09-0.96, P = 0.043). Higher albumin levels were associated with a 4% decrease in the odds of anemia (OR = 0.96, 95% CI: 0.92-0.99, P = 0.036). Females on NNRTI regimens had a 133% increase in the odds of anemia compared to those on INSTI regimens (OR = 2.33, 95% CI: 1.32-4.09, P = 0.003). Microcytosis was associated with a 107% increase in the odds of anemia (OR = 2.07, 95% CI: 1.04-4.09, P = 0.038). Lastly, hypertension was associated with a 57% decrease in the odds of anemia (OR = 0.43, 95% CI: 0.22-0.84, P = 0.013).

In the adjusted analysis, lower waist circumference was associated with a 3% increase in the odds of anemia (OR = 0.97, 95% CI: 0.95-0.99, P = 0.018), Table 2. Higher albumin levels were linked to a 4% decrease in the odds of anemia (OR = 0.96, 95% CI: 0.92-0.99, P = 0.047). Females on NNRTI regimens had a 2.78 times higher odds of anemia compared to those on INSTI regimens (OR = 2.78, 95% CI: 1.33-5.82, P = 0.006). Microcytosis was associated with a 3.18 times higher odds of anemia (OR = 3.18, 95% CI: 1.27-7.99, P = 0.014). Lastly, hypertension was associated with a 66% decrease in the odds of anemia (OR = 0.34, 95% CI: 0.14-0.87, P = 0.024).

## Factors associated with anemia in males

In the unadjusted analysis, several factors were significantly associated with anemia in men (Table 3). A higher BMI was associated with an 8% decrease in the odds of anemia (OR = 0.92, 95% CI: 0.85-0.99, P = 0.034). Lower waist circumference was associated with a 3% increase in the odds of anemia (OR = 0.97, 95% CI: 0.95-0.99, P = 0.014). Men on NNRTI regimens had a 194% increase in the odds of anemia compared to those on INSTI regimens (OR = 2.94, 95% CI: 1.29-6.70, P = 0.01). Higher creatinine levels were associated with a 1% decrease in the odds of anemia (OR = 0.99, 95% CI: 0.96-0.99, P = 0.014). Higher albumin levels were associated with a 5% decrease in the odds of anemia (OR = 0.95, 95% CI: 0.91-0.99, P = 0.019).

In the adjusted analysis, no variables remained significantly associated with anemia in men.

## Discussion

People Living with HIV (PLWH) who are on combinational ART still experience anemia, but the factors contributing to its persistence are not well understood in Zambia. People living with HIV (PLWH) on combination ART still experience anemia, but the associated factors are not well understood. In this study, we examined the sex differences in factors associated with anemia in an adult population of 631 PLWH in, Zambia. We found a significantly higher prevalence of anemia in females compared to males. Among females, waist circumference, albumin levels, being an NNRTI based regimen and microcytosis was also associated with anemia in this cohort but this relationship was abrogated by male sex as no variable remained

**Table 3. Logistic regression analysis of factors associated with anemia in males.**

| Variables | Unadjusted analysis | | Adjusted Analysis | |
|---|---|---|---|---|
| | OR (95% CI) | Pvalue | AOR (95% CI) | pvalue |
| Age (years) | 1.02 (0.99,1.04) | 0.088 | 1.01 (0.97-1.05) | 0.656 |
| Bmi kg/m$^2$ | 0.92 (0.85,0.99) | **0.034** | 0.97 (0.85-1.11) | 0.666 |
| Waist circumference (cm) | 0.97 (0.95,.99) | **0.014** | 0.97 (0.93-1.01) | 0.111 |
| Duration on ART mo, m(IQR) | 1.00 (0.99, 1.00) | 0.391 | | |
| ART regimen | | | | |
| *INSTI* | 1 | | | |
| *NNRTI* | 2.94 (1.29,6.70) | **0.01** | 2.59 (0.59-11.46) | 0.21 |
| *PI* | 2.23 (0.74, 6.70) | 0.154 | 2.53 (0.39 -16.08) | 0.324 |
| Types of Anemia | | | | |
| *Microcytosis* | 0.30 (0.09-0.99) | 0.050 | 0.31 (0.04-2.46) | 0.268 |
| *Normocytic* | 0.47 (0.24-0.91) | **0.026** | 0.64 (0.21-1.93) | 0.426 |
| CD4+ count | | | | |
| *>350* | 1 | | | 0.411 |
| *201-350* | 0.36 (0.12-1.06) | 0.064 | 0.38 (0.07-2.09) | 0.267 |
| *<200* | 0.72 (0.19-2.72 | 0.628 | 0.34 (0.03-4.19) | 0.401 |
| Lymphocytes 10$^9$/L | 1.34 (0.90, 1.98) | 0.147 | | |
| Monocytes 10$^9$/L | 0.58 (0.17, 1.99) | 0.39 | | |
| Creatinine μmol/l | 0.99 (0.96, 0.99) | **0.014** | 0.98 (0.95-1.01) | 0.114 |
| Urea mg/dl | 0.72 (0.55-0.94) | **0.016** | 0.77 (0.47-1.26) | 0.296 |
| Albumin g/dl | 0.95 (0.91, 0.99) | **0.019** | 1.03 (0.96-1.09) | 0.479 |

Abbreviations: BMI Body Mass Index, Alt Alanine Aminotransferase, WBC White cell count, mo months, cm centre meters, mmHg millimetres of mercury, HTN HypertensionART Antiretroviral therapy

statistically significant on multivariable analysis. Our findings suggest the need to closely monitor anemia among PLWH, especially among females.

Our study found an overall prevalence of anemia among PLWH of 36%, which is higher than the 16.8% reported in Uganda [29], 25.35% in East Africa [30], and 34% in Ethiopia [31] but lower than that reported in other studies including Indonesia 40.3%[32], China 55.15%[33], Nepal 66.7%[34]. Additioanlly, a global systematic review and meta-analysis found a prevalence of 46.6% for adults PLWH[2]. Our findings and those reported in other studies demonstrate the need to effectively monitor the haemoglobin levels of PLWH.

The sex differences observed may be due to several factors: biological differences such as menstrual blood loss and hormonal influences, greater nutritional deficiencies among women in low-resource settings, socio-economic barriers limiting women's access to healthcare and nutrition [35,36]. The reduced odds of anemia with increasing age may partly reflect hormonal changes in females, particularly the cessation of menstruation during menopause, which reduces blood loss and improves iron stores [37]. However, these findings were not significant in adjusted analyses, indicating the potential role of confounding factors.

Among females, microcytosis is linked to iron deficiency due to menstrual blood loss, exacerbated by chronic inflammation of HIV [38].

Although the exact mechanisms are not fully understood, the association of Anemia with lower blood pressure maybe explained by several biological mechanisms that have been suggested. Anemia reduces the free haemoglobin's binding to nitric oxide, leading to vasodilation and lower blood pressure [39,40]. Additionally, previous reports have indicated that haemoglobin levels is strongly related to arterial stiffness, as measured by pulse wave velocity, which inversely predisposes these individuals to hypotension [41]. Lastly, chronic inflammation and immune activation in HIV infection can also lead to vasodilation and may impair iron availability for erythropoiesis, exacerbating anemia [42].

In this study, lower waist circumference was associated with a higher risk of anemia in PLWH. A lower waist circumference may reflect changes in body composition, such as reduced muscle mass or fat stores, which is associated with anemia through unknown mechanisms [43]. Studies have shown that anemia is associated with low waist circumference in HIV-infected women [44]. A lower waist circumference in females may reflect underlying nutritional deficiencies or higher levels of chronic inflammation, both of which are known contributors to anemia in PLWH [36]. Additionally, low albumin levels, associated with anemia in people living with HIV, also reflect poor nutritional status and increased catabolism [45]. Albumin, is also a negative acute-phase reactant, decreases during inflammatory states, further suppressing erythropoiesis [46]. Addressing nutritional deficiencies and managing inflammation are critical in mitigating anemia in this population.

NNRTIs were associated with higher anemia risk compared to INSTIs. The exact mechanisms underlying this association are not known but the impact of NNRTI maybe related to long term use, potentially leading bone toxicities and haematological complications, since most of our participants were initiated on NNRTIs [47]. Our findings differ from Harding et al., who reported a higher anemia risk in PLWH on INSTI regimens compared to those on NNRTIs. [48].

The study's findings emphasize the necessity for sex-specific interventions to manage anemia among PLWH.. For females, interventions should focus on addressing nutritional suppport. Additionally, transitioning women from NNRTI-based regimens to alternatives like INSTIs when appropriate can reduce the risk of anemia. Regular monitoring of haemoglobin levels, waist circumference, and albumin levels in women, along with comprehensive reproductive health services, will help alleviate anemia and improve overall health outcomes.

Future research should include larger and more diverse sample sizes to enhance generalizability, and investigate contextual factors such as diet, socioeconomic status, and healthcare access, which impact anemia across regions and sexes. Longitudinal studies are needed to explore the effects of ART regimens, nutritional status, and anemia progression. Incorporating advanced biomarkers for nutritional anemia, inflammation, and immune activation will help understand underlying mechanisms, while exploring hormonal and genetic influences on sex-specific anemia risk could lead to more targeted management strategies. Furthermore, future studies should explore anemia prevalence and associated factors in HIV-negative individuals from the same population to provide comparative insights Our study's limitations include the absence of direct measurements of markers of nutritional anemia and inflammatory markers, as well as missing data on a few variables, potentially affecting our analysis. Despite this our study reveals key factors linked to anemia in PLWH, emphasizing sex-specific differences

## Conclusions

Our study revealed a higher prevalence of anemia in females of 41% compared to males 27% in PLWH including the severe forms of anemia, i.e., moderate and severe anemia. Associated factors differed by sex including blood pressure, waist circumference, microcytosis and NNRTI based regimen were associated with anemia in females but this relationship was abrogated by male sex. These findings emphasize the importance of sex-specific strategies in managing anemia among PLWH.

## Supporting information

**S1 File. The strobe checklist.**
(PDF)

**S2 File. Minimal dataset.**
(XLSX)

## Author contributions

**Conceptualization:** Kingsley Kamvuma, Sepiso K Masenga, Sody M. Munsaka.

**Data curation:** Benson M Hamooya, Yusuf Uthman Ademola.

**Formal analysis:** Kingsley Kamvuma, Benson M Hamooya, Steward Mudenda, Sepiso K Masenga, Sody M. Munsaka.

**Investigation:** Kingsley Kamvuma, Benson M Hamooya, Sody M. Munsaka.

**Methodology:** Kingsley Kamvuma, Yusuf Uthman Ademola, Alfred Machiko, Sepiso K Masenga, Sody M. Munsaka.

**Project administration:** Kaseya O.R. Chiyenu, Alfred Machiko, Sepiso K Masenga.

**Software:** Kingsley Kamvuma.

**Supervision:** Kingsley Kamvuma, Benson M Hamooya, Kaseya O.R. Chiyenu, Sepiso K Masenga, Sody M. Munsaka.

**Validation:** Yusuf Uthman Ademola, Sody M. Munsaka.

**Visualization:** Kingsley Kamvuma, Kaseya O.R. Chiyenu, Steward Mudenda, Alfred Machiko, Sody M. Munsaka.

**Writing – original draft:** Kingsley Kamvuma.

**Writing – review & editing:** Kaseya O.R. Chiyenu, Yusuf Uthman Ademola, Steward Mudenda, Alfred Machiko, Sepiso K Masenga, Sody M. Munsaka.

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
