## [Decision Letter · Decision Letter 0]

14 Jan 2025

PONE-D-24-48707Sex Differences in the Risk Profiles for Anaemia in People Living with HIV, A Cross Sectional StudyPLOS ONE

Dear Dr. Kamvuma,

Thank you for submitting your manuscript to PLOS ONE. After careful consideration, we feel that it has merit but does not fully meet PLOS ONE’s publication criteria as it currently stands. Therefore, we invite you to submit a revised version of the manuscript that addresses the points raised during the review process.

Please submit your revised manuscript by Feb 28 2025 11:59PM. If you will need more time than this to complete your revisions, please reply to this message or contact the journal office at plosone@plos.org . Please include the following items when submitting your revised manuscript:

We look forward to receiving your revised manuscript.

Kind regards,

Zivanai Cuthbert Chapanduka, MBChB (M.D)

Academic Editor

PLOS ONE

Journal requirements: When submitting your revision, we need you to address these additional requirements. 1. Please ensure that your manuscript meets PLOS ONE's style requirements, including those for file naming. The PLOS ONE style templates can be found at https://journals.plos.org/plosone/s/file?id=wjVg/PLOSOne_formatting_sample_main_body.pdf and https://journals.plos.org/plosone/s/file?id=ba62/PLOSOne_formatting_sample_title_authors_affiliations.pdf.

Additional Editor Comments:

Dear Dr Kavuma

Thank you for the submission.

Both peer reviewers require minor revisions to your manuscript. I shall send the manuscript back to you shortly. Kindly attend to these carefully ensuring that every aspect is satisfactorily attended to. Please note that the second round of review is the final one.

Kind regards

Reviewers' comments:

Reviewer's Responses to Questions

**Comments to the Author**

1. Is the manuscript technically sound, and do the data support the conclusions?

Reviewer #1: Yes

Reviewer #2: Yes

2. Has the statistical analysis been performed appropriately and rigorously? 

Reviewer #1: Yes

Reviewer #2: Yes

3. Have the authors made all data underlying the findings in their manuscript fully available?

Reviewer #1: Yes

Reviewer #2: Yes

4. Is the manuscript presented in an intelligible fashion and written in standard English?

Reviewer #1: Yes

Reviewer #2: Yes

5. Review Comments to the Author

Reviewer #1: The paper highlights important issues in management of people living with HIV (PLWH) and has some novel approach hence I believe with some corrections it will be ready for publication.

There are several spelling mistakes that need to be addressed. There is also a mixture of American and British English (hemoglobin vs haemoglobin for example). There are several statements that need to be rephrased.

To add weight to the manuscript, I recommend a thorough discussion of all findings, especially unusual findings, based currently available evidence.

Some of the issues are highlighted on the attached manuscript.

It would be interesting to know the rates of anemia and associated factors in the people without HIV in the same population from which the sample was taken. (not a requirement).

Reviewer #2: The study presents a valuable examination of anaemia among PLWH, revealing significant sex differences in prevalence and associated risk factors. By addressing identified suggested improvements, the overall clarity, relevance, and impact of the research will be significantly enhanced, further contributing to the body of knowledge in this critical area of public health.

6. PLOS authors have the option to publish the peer review history of their article (what does this mean? ). If published, this will include your full peer review and any attached files.

**Do you want your identity to be public for this peer review?** For information about this choice, including consent withdrawal, please see our Privacy Policy .

Reviewer #1: **Yes: ** Leonard Mutema

Reviewer #2: **Yes: ** Ernest MUSEKWA

---

## [Author Response · Author response to Decision Letter 0]

18 Jan 2025

Point-by-Point Response to Reviewer Comments

REVIEWER 1

Response to Reviewer’s Comments

We greatly appreciate the positive feedback and the recognition of the importance of the findings in our manuscript. We have carefully considered all review comments and appropriately addressed each of them in the revised manuscript

Spelling and Language Consistency:

• We have addressed the spelling mistakes throughout the manuscript and ensured consistency in language usage and revised incorrect statements.

Discussion of Findings:

• We have expanded the discussion of all findings, particularly focusing on the unusual results, as suggested (See line 316 to 319). This includes a broader exploration of why certain findings, may be unusual and how they fit within the broader context of our study population ( line 331 to 337).

Rates of Anaemia in People Without HIV:

• While not a requirement, we acknowledge the suggestion to include data on anaemia and associated factors in the general population (without HIV). While this was not part of the current study design, we included a note in the discussion (line 367 to 369) on the potential value of future studies comparing anaemia rates between PLWH.

REVIEWER 2

Background:

1. Comment: Add more detail on the relationship between HIV and anaemia.

o Response: We expanded the discussion on HIV and anaemia (line 77 to 86), highlighting HIV-induced chronic inflammation and ART-related factors contributing to anaemia.

2. Comment: Provide more information on investigating sex differences in anaemia among PLWH.

o Response: We included a detailed explanation of the importance of sex differences (line 92 to 97).

Methods: a) Comment: Brief overview of questionnaires and data collection.

• Response: We clarified the data collection methods (line 123 to 128), mentioning the use of standardized questionnaires and clinical data from ART registries.

b) Comment: Clarify rationale for including only adults and minimum treatment duration.

• Response: We clarified that adults were included due to physiological differences (line 111 to 116) and the six-month ART duration ensures stable treatment effects.

Results: a) Comment: Add clarity in statistical interpretations.

• Response: We revised and added further discussion (line 331 to 337), emphasizing the clinical significance of waist circumference and ART regimen associations with anaemia.

b) Comment: Clarify the comparison between males and females regarding waist circumference.

• Response: The sentence was revised for clarity (line 220 to 223).

Discussion: a) Comment: Avoid generalizing to all of Sub-Saharan Africa.

• Response: We revised the manuscript to avoid generalisation to the Sub-saharan africa

b) Comment: Outline areas for future research.

• Response: Future research directions were added (line 359 to 366), suggesting larger sample sizes and further exploration of contextual factors like socio-economic status and nutrition.

We thank the reviewers for their valuable feedback, which has helped improve the manuscript.

---

## [Decision Letter · Decision Letter 1]

5 Feb 2025

Sex Differences in the Risk Profiles for Anaemia in People Living with HIV, A Cross Sectional Study

PONE-D-24-48707R1

Dear Dr. Kamvuma

We’re pleased to inform you that your manuscript has been judged scientifically suitable for publication and will be formally accepted for publication once it meets all outstanding technical requirements.

Kind regards,

Zivanai Cuthbert Chapanduka, MBChB (M.D)

Academic Editor

PLOS ONE

Additional Editor Comments (optional):

Reviewers' comments:

Reviewer's Responses to Questions

**Comments to the Author**

1. If the authors have adequately addressed your comments raised in a previous round of review and you feel that this manuscript is now acceptable for publication, you may indicate that here to bypass the “Comments to the Author” section, enter your conflict of interest statement in the “Confidential to Editor” section, and submit your "Accept" recommendation.

Reviewer #1: All comments have been addressed

Reviewer #2: All comments have been addressed

2. Is the manuscript technically sound, and do the data support the conclusions?

Reviewer #1: Yes

Reviewer #2: Yes

3. Has the statistical analysis been performed appropriately and rigorously? 

Reviewer #1: Yes

Reviewer #2: Yes

4. Have the authors made all data underlying the findings in their manuscript fully available?

Reviewer #1: Yes

Reviewer #2: Yes

5. Is the manuscript presented in an intelligible fashion and written in standard English?

Reviewer #1: Yes

Reviewer #2: Yes

6. Review Comments to the Author

Reviewer #1: There are a few small errors and suggestions highlighted on the uploaded manuscript. The manuscript should therefore be accepted provided these are addressed.

Reviewer #2: Using "anaemia" or "anemia" interchangeably can lead to confusion, inconsistency, or miscommunication.

It is best to choose one term based on the relevant linguistic, contextual, and professional standards.

The following are important reasons for not using them interchangeably in a text:

1. Regional Variations:

"Anaemia" is the preferred spelling in British English, while "anemia" is the standard in American English. Using the appropriate spelling for the target audience is important for clarity and professionalism.

2. Consistency:

Maintaining consistent terminology throughout a text enhances readability and avoids confusion for the reader. Switching between "anaemia" and "anemia" could lead to misunderstanding or misinterpretation.

3. Academic Standards:

Many academic and professional publications have specific guidelines regarding language usage, including spelling conventions. Adhering to these standards is essential for scientific communication and credibility.

4. Audience Consideration:

Depending on the audience, using one term over the other may resonate better. For example, if the audience is primarily from a British context, "anaemia" would be more recognizable and appropriate.

5. Professional Context:

In clinical practice, using uniform terminology is essential for effective communication among healthcare professionals. Variability in spelling could lead to miscommunication in patient records, research papers, or treatment protocols.

7. PLOS authors have the option to publish the peer review history of their article (what does this mean? ). If published, this will include your full peer review and any attached files.

**Do you want your identity to be public for this peer review?** For information about this choice, including consent withdrawal, please see our Privacy Policy .

Reviewer #1: **Yes: ** Leonard Mutema

Reviewer #2: **Yes: ** EM MUSEKWA

---

## [Editor Report · Acceptance letter]

PONE-D-24-48707R1

PLOS ONE

Dear Dr. Kamvuma,

I'm pleased to inform you that your manuscript has been deemed suitable for publication in PLOS ONE. Congratulations! Your manuscript is now being handed over to our production team.

Kind regards,

on behalf of

Professor Zivanai Cuthbert Chapanduka

Academic Editor

PLOS ONE